# SymRAG: Efficient Neuro-Symbolic Retrieval Through Adaptive Query Routing

**Safayat Bin Hakim**                                          SHAKIM3@UMBC.EDU
*Department of Information Systems, University of Maryland, Baltimore County, USA*

**Muhammad Adil**                                          MUHAMMAD.ADIL@IEEE.ORG
*Department of Computer Science and Engineering, University at Buffalo, USA*

**Alvaro Velasquez**                                          ALVARO.VELASQUEZ@COLORADO.EDU
*Department of Computer Science, University of Colorado Boulder, USA*

**Houbing Herbert Song**                                          H.SONG@IEEE.ORG
*Department of Information Systems, University of Maryland, Baltimore County, USA*

**Editors:** Leilani H. Gilpin, Eleonora Giunchiglia, Pascal Hitzler, and Emile van Krieken

## Abstract

Current Retrieval-Augmented Generation systems use uniform processing, causing inefficiency as simple queries consume resources similar to complex multi-hop tasks. We present SymRAG, a framework that introduces adaptive query routing via real-time complexity and load assessment to select symbolic, neural, or hybrid pathways. SymRAG's neuro-symbolic approach adjusts computational pathways based on both query characteristics and system load, enabling efficient resource allocation across diverse query types. By combining linguistic and structural query properties with system load metrics, SymRAG allocates resources proportional to reasoning requirements. Evaluated on 2,000 queries across HotpotQA (multi-hop reasoning) and DROP (discrete reasoning) using Llama-3.2-3B and Mistral-7B models, SymRAG achieves competitive accuracy (97.6–100.0% exact match) with efficient resource utilization (3.6–6.2% CPU utilization, 0.985–3.165s processing). Disabling adaptive routing increases processing time by 169–1151%, showing its significance for complex models. These results suggest adaptive computation strategies are more sustainable and scalable for hybrid AI systems that use dynamic routing and neuro-symbolic frameworks.

## 1. Introduction

Retrieval-Augmented Generation (RAG) systems address factual inconsistencies in Large Language Models by grounding generation in external knowledge, yet they face a fundamental efficiency problem: simple queries consume computational resources equivalent to complex multi-hop reasoning tasks (Fan et al., 2024; Zhao et al., 2024). Current RAG architectures apply uniform processing pipelines regardless of query complexity, leading to substantial computational overhead from dense neural retrieval and quadratic attention scaling (Chitty-Venkata et al., 2024). This one-size-fits-all approach becomes increasingly problematic as model scale and deployment demands grow, particularly given rising concerns about AI's energy footprint and the need for sustainable deployment.

We present SymRAG, a neuro-symbolic approach that introduces adaptive query routing to match computational intensity with reasoning requirements while maintaining robustness across diverse query types. SymRAG's main contribution lies in real-time complexity

assessment that dynamically routes queries to symbolic, neural, or hybrid pathways based on both query characteristics and system load. This approach combines symbolic pre-filtering to reduce neural computation burden with rule-based reasoning for logical consistency, enabling efficient resource allocation across diverse query types.

Our key contributions focus on three important aspects to improve adaptive RAG.

- **Adaptive query routing**: Dynamic pathway selection based on real-time complexity and load assessment, with specialized symbolic-neural fusion for different reasoning types.
- **Resource-aware processing**: Computational efficiency maintaining CPU usage below 6.2% while achieving competitive accuracy (97.6-100% exact match).
- **Cross-architecture validation**: Empirical validation across Llama-3.2-3B (Llama, 2024) and Mistral-7B (MistralAI, 2024), demonstrating that disabling adaptive logic causes 169-1151% processing time increases.

Evaluation on 2,000 queries from HotpotQA and DROP datasets demonstrates SymRAG's effectiveness, achieving 100% success on HotpotQA and 97.6% on DROP with efficient resource utilization. The system exhibits adaptive behavior, favoring neural (64% HotpotQA) and hybrid (60.2% DROP) pathways over pure symbolic reasoning, with cross-model validation confirming computational benefits across different architectures.

## 2. Related Work

Retrieval-Augmented Generation enriches LLMs with external knowledge (Fan et al., 2024; Zhao et al., 2024) but often requires trade-offs between high computational cost (neural-only) and brittle logic (symbolic-only). Hybrid RAG methods attempt to bridge this gap through various integration strategies.

Several hybrid approaches incorporate symbolic knowledge but lack dynamic adaptation to query complexity or system resources. RuleRAG (Chen et al., 2024) and RuAG (Zhang et al., 2024) inject handcrafted or learned rules into retrieval and generation pipelines, while HybridRAG (Sarmah et al., 2024) statically fuses knowledge graphs with vector contexts without considering query-specific requirements or computational constraints.

Another line of work focuses on graph-based reasoning approaches. Graph-based methods like CRP-RAG (Xu et al., 2024) construct reasoning graphs for multi-hop QA but rely on multiple sequential LLM calls and fixed structures. CDF-RAG (Khatibi et al., 2025) adds causal graphs with reinforcement learning-driven query refinement, but its RL loop introduces approximately 100ms latency per query without optimizing overall system throughput.

SymRAG differs fundamentally by implementing real-time adaptive routing based on complexity assessment and resource monitoring. Unlike static integration approaches, SymRAG dynamically selects optimal processing pathways (symbolic, neural, or hybrid) for each query, achieving high accuracy (97.6-100% exact match) with efficient resource utilization explicitly reported across multiple metrics. A detailed comparison of RAG frameworks emphasizing SymRAG's unique contributions is provided in Appendix A.11.

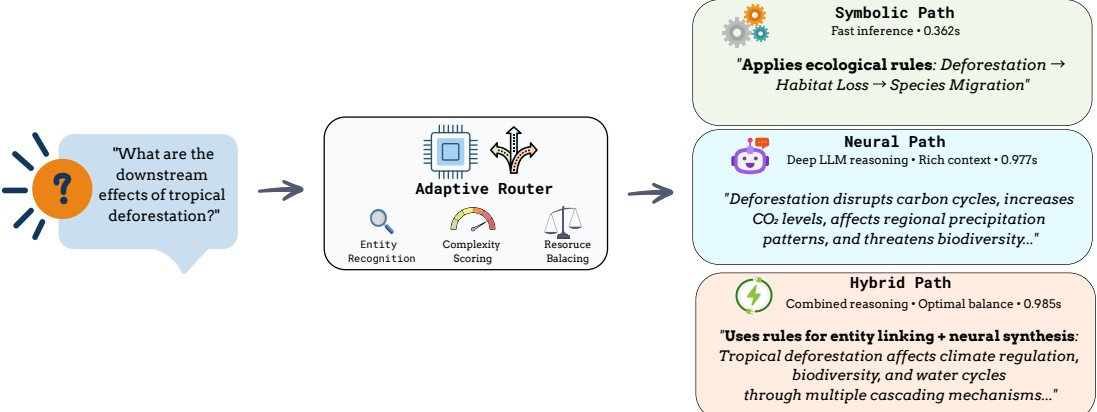

Figure 1: SymRAG adaptive query routing framework showing adaptive selection for tropical deforestation query. The system chose hybrid processing (0.985s) over faster alternatives based on query complexity and resource state, demonstrating intelligent trade-offs between speed, accuracy, and system load.

## 3. SymRAG: Adaptive Query Routing Framework

SymRAG introduces a novel adaptive query processing pipeline that dynamically routes queries through specialized pathways based on real-time complexity assessment and system resource monitoring. Figure 1 illustrates the framework's core innovation: intelligent routing between symbolic, neural, and hybrid processing paths, demonstrating emergent adaptive behavior where the system discovers optimal strategies beyond initial design assumptions.

**Architectural Overview:** SymRAG integrates four core components—Graph-Based Symbolic Reasoner, Neural Retriever, Hybrid Integrator, and Resource-Aware System Control Manager—operating across Query Processing, Hybrid Processing, and Resource Management stages. The system employs utility-based path selection that balances accuracy, latency, and computational costs through continuous learning and threshold adaptation. Unlike static integration approaches, SymRAG exhibits meta-learning characteristics where optimal resource allocation patterns emerge through experience, enabling both superior performance and computational sustainability. Complete architectural details, component interactions, and system implementation specifics are provided in Appendix A.2.

The framework operates through three integrated stages where the SystemControlManager coordinates intelligent routing decisions by leveraging insights from the QueryExpander and ResourceManager working in tandem. The QueryExpander analyzes input query $q$ to compute a complexity score $\kappa$ combining linguistic and structural properties that capture both general and specialized complexity indicators, while the ResourceManager continuously monitors four key metrics—CPU$(t)$, GPU$(t)$, MEM$(t)$, and Power$(t)$—to form the state vector $R(t)$, enabling dynamic adaptation of system behavior based on real-time computational load.

**Definition 1 (Query Complexity $\kappa(q)$)** *The complexity $\kappa(q)$ of query $q$ combines linguistic and structural properties:*

$$\kappa(q) = (w_A \cdot A(q) + w_L \cdot L(q)) \cdot (1 + S_H(q))$$

where $A(q)$ represents mean attention values from a pre-trained language model capturing cognitive processing effort, $L(q)$ is normalized query length, $S_H(q) = w_{sh1} \cdot N_{ents}(q) + w_{sh2} \cdot N_{hops}(q)$ incorporates structural heuristics for named entities and multi-hop indicators, and $w_A, w_L, w_{sh1}, w_{sh2}$ are empirically determined weights.

For datasets involving structured reasoning like DROP, pattern-based rules provide direct processing path suggestions by matching queries against predefined constraints, creating an effective complexity score $\kappa_{eff}(q)$ that combines general assessment with domain-specific indicators.

**Definition 2 (Path Selection Policy)** *The adaptive path selection policy $\pi$ maps query complexity $\kappa_{eff}(q)$ and system resource state $R(t)$ to optimal path $P^*$ through utility maximization:*

$$P^* = \pi(\kappa_{eff}(q), R(t)) = \arg \max_{P \in \{P_S, P_N, P_H\}} \mathcal{U}(P|\kappa_{eff}(q), R(t))$$

*where the utility function balances accuracy, latency, and resource costs:*

$$\mathcal{U}(P|\kappa_{eff}, R) = w_{acc} \cdot E[Acc(P, q)] - w_{lat} \cdot E[Lat(P, q)] - w_{cost} \cdot E[Cost(P, q, R)]$$

The algorithm utilizes dynamically adjusted thresholds $T_{sym}$ and $T_{neural}$ to route queries. The adaptive mechanism responds effectively to varying system loads. Under high GPU utilization, the system raises $T_{neural}$ to route more queries to hybrid or symbolic paths. During low-pressure periods, thresholds adjust to leverage available neural processing capacity. This threshold adaptation enables meta-learning characteristics where the system discovers optimal allocation patterns through experience, evidenced by CPU usage decreasing over time as increasingly effective resource management strategies emerge.

Experimental findings reveal striking emergent behavior that challenges conventional neuro-symbolic assumptions. The system rarely selects pure symbolic paths (0.1% for DROP, 0% for HotpotQA), suggesting modern LLMs have internalized substantial symbolic reasoning capabilities. Neural paths handle 41.7% of DROP and 64% of HotpotQA queries, while hybrid paths process 60.2% of DROP and 36% of HotpotQA queries, demonstrating the system's learned preference for approaches that maximize empirical effectiveness.

SymRAG employs three distinct integration strategies optimized for different reasoning requirements. The *symbolic-only path* processes queries entirely through the GraphSymbolicReasoner using knowledge graph-based approaches with predefined rules and symbolic inference, achieving exceptional speed (0.362s) but limited accuracy (31.6% on DROP) when used in isolation. The *neural-only path* handles queries through the NeuralRetriever using dense retrieval techniques followed by LLM-based generation, achieving strong independent performance that validates the system's preference for neural processing. The *hybrid path* represents SymRAG's most sophisticated approach, strategically combining symbolic and neural processing through the HybridIntegrator using specialized fusion mechanisms: HotpotQA employs embedding alignment for multi-hop reasoning that creates unified semantic spaces through cross-modal attention, while DROP uses structured reconciliation for discrete answer types with type agreement protocols and value reconciliation strategies, ensuring synergy across diverse reasoning tasks.

SymRAG's design embodies key principles from neuro-symbolic AI theory while introducing novel concepts around adaptive resource management. The system demonstrates characteristics of loosely-coupled hybrid integration with bidirectional information flow, where symbolic components guide neural retrieval through scoring adjustments while neural confidence scores influence fusion decisions. Unlike systems requiring extensive pre-encoded symbolic knowledge, SymRAG dynamically extracts rules from data, as demonstrated by DROP experiments with 533 dynamically extracted rules. This adaptive knowledge acquisition bridges the gap between rigid symbolic systems and purely learned approaches, pioneering resource-aware neuro-symbolic computing by explicitly modeling reasoning-resource trade-offs aligned with sustainable AI principles. The implementation of SymRAG, including code and configurations, is available at https://github.com/sbhakim/symrag.git.

## 4. Experimental Setup

We conducted extensive experiments on two complementary question-answering datasets to rigorously evaluate SymRAG's performance and validate our design decisions across multiple LLM architectures. Our framework was evaluated on Llama-3.2-3B and Mistral-7B-Instruct-v0.3 to assess generalizability, with methodology emphasizing statistical validity through large-scale evaluation and comprehensive ablation studies.

We selected datasets testing fundamentally different reasoning capabilities to assess SymRAG's versatility and adaptive behavior. HotpotQA (Yang et al., 2018) provides a challenging multi-hop question answering benchmark requiring reasoning across multiple documents, with 1,000 development set queries demanding integration of information from multiple supporting facts while testing complex inference chains and handling textual ambiguity. DROP (Dua et al., 2019) offers a discrete reasoning benchmark evaluating numerical operations, comparisons, and span extraction over paragraphs, with 1,000 queries providing robust statistical power for discrete mathematical operations requiring precise understanding and manipulation of quantities, dates, and entities.

We employ comprehensive metrics evaluating both effectiveness and efficiency, recognizing that practical deployment requires balancing accuracy with computational resources. Success rates use Exact Match and F1-score for HotpotQA and exact correctness percentages for DROP structured answers. Processing times include average, median, and 95th percentile measurements per query. Resource utilization monitoring captures CPU, memory, and GPU usage at 100ms intervals throughout query processing, with energy efficiency calculations normalizing consumption units as functions of resource utilization and processing time. Statistical significance employs appropriate tests with p-values and effect sizes to quantify practical significance.

Primary baselines include Neural-Only RAG using solely the NeuralRetriever component representing current state-of-the-art approaches, and Symbolic-Only processing exclusively through the GraphSymbolicReasoner demonstrating pure rule-based reasoning capabilities. Ablation configurations systematically remove or modify key components: disabling adaptive logic forces all queries through hybrid paths regardless of complexity or resource state, removing few-shot prompting assesses impact on discrete reasoning tasks, and comparing static versus dynamic rules evaluates automated rule learning benefits. Complete experi-

Table 1: Cross-Model Performance: SymRAG on Llama-3.2-3B and Mistral-7B-Instruct

| Model | Dataset | Exact Match Rate (%) | Avg Time (sec) | Resource Utilization (%) | | | Path Distribution (S/N/H) (%) |
|-------|---------|-----|-----|-----|--------|-----|-----|
| | | | | CPU | Memory | GPU | |
| Llama-3.2-3B | DROP | **99.4** | $0.985 \pm 1.29$ | 4.6 | 5.6 | 41.1 | 0.1 / 41.7 / 60.2 |
| | HotpotQA | **100.0** | $1.991 \pm 1.67$ | 6.2 | 5.6 | 42.8 | 0.0 / 64.0 / 36.0 |
| Mistral-7B | DROP | **97.6** | $2.443 \pm 3.45$ | 3.6 | 8.0 | 66.0 | 0.0 / 43.6 / 56.4 |
| | HotpotQA | **100.0** | $3.165 \pm 2.83$ | 3.9 | 10.8 | 68.2 | 0.0 / 64.0 / 36.0 |

S/N/H: Symbolic/Neural/Hybrid path percentages.

mental setup details including hardware specifications and hyperparameters are provided in Appendix A.10.

## 5. Results and Analysis

This section presents comprehensive analysis of SymRAG's performance through extensive experimentation on 1,000 queries per dataset across two LLM architectures, demonstrating both expected behaviors and adaptive patterns that provide insights about system behavior and the evolving role of symbolic reasoning in modern AI systems.

### 5.1. Overall System Performance

Table 1 summarizes SymRAG's performance across both datasets and LLM architectures, presenting key insights about the system's adaptive behavior and generalizability that challenge conventional assumptions about neuro-symbolic integration.

SymRAG demonstrates high accuracy across both architectures, with Llama-3.2-3B achieving 100% exact match on HotpotQA and 99.4% on DROP, while Mistral-7B achieves 100% exact match on HotpotQA and 97.6% on DROP. The single failure case in DROP involved complex multi-step calculation with ambiguous temporal references, highlighting areas for future improvement. Processing times show architectural differences: Llama-3.2-3B processes queries in 0.985s (DROP) and 1.991s (HotpotQA), while Mistral-7B requires 2.443s (DROP) and 3.165s (HotpotQA), reflecting the larger model's computational requirements but maintaining acceptable response times.

Resource utilization patterns demonstrate architectural trade-offs. Llama-3.2-3B maintains minimal CPU usage (4.6-6.2%) with moderate GPU utilization (41-43%), while Mistral-7B shows lower CPU usage (3.6-3.9%) but higher GPU utilization (66-68.2%) and memory usage (8.0-10.8%), reflecting different optimization strategies. Both models maintain stable memory usage, demonstrating effective resource management across architectures.

Most significantly, path distributions show consistent preferences across models. Both Llama-3.2-3B and Mistral-7B demonstrate near-zero usage of pure symbolic paths and strong preference for hybrid processing for DROP (60.2% and 56.4% respectively). For HotpotQA, both models converge on an identical path distribution (64% neural, 36% hybrid), validating the emergent adaptive behavior findings across different LLM architectures. This consistency suggests that modern language models have indeed internalized substan-

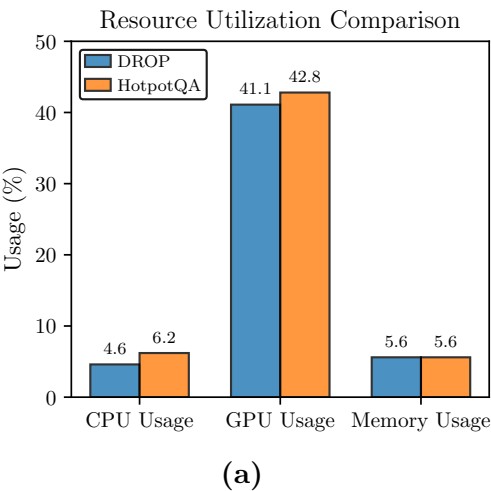
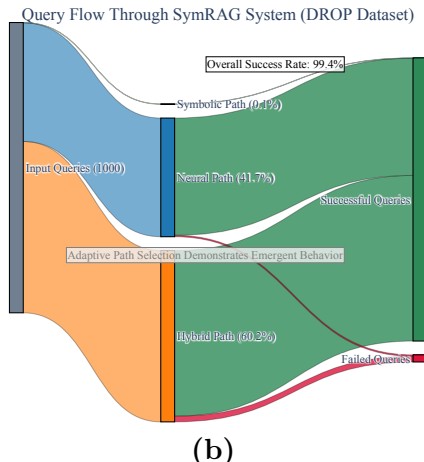

**(a)**

**(b)**

Figure 2: SymRAG system performance: (a) Resource utilization across DROP and Hot-potQA datasets showing efficient operation with CPU usage below 6.2%, and (b) Query flow distribution demonstrating emergent adaptive behavior with 60.2% hybrid processing, challenging conventional fixed neuro-symbolic routing assumptions.

Table 2: Cross-Model Impact of Disabling Adaptive Logic

| Model | Configuration | EM (%) | F1 (%) | Avg Time (s) | Acc. Impact | Speed Impact | p-value |
|---|---|---|---|---|---|---|---|
| | With Adaptive | 89.2 | 89.4 | 0.985 | — | — | — |
| Llama-3.2-3B | No Adaptive | 75.7 | 75.9 | 2.645 | –15.1% | **+168.6%** | <0.001 |
| | *Effect Size (d)* | *0.78* | *0.78* | *—* | *Large* | *Large* | *—* |
| | With Adaptive | 89.2 | 89.4 | 2.443 | — | — | — |
| Mistral-7B | No Adaptive | 86.2 | 86.1 | 25.847 | –3.4% | **+958.0%** | <0.001 |
| | *Effect Size (d)* | *0.03* | *0.03* | *–0.60* | *Small* | *Very Large* | *—* |

tial symbolic reasoning capabilities, making hybrid integration more effective than pure symbolic approaches.

As illustrated in Figure 2, SymRAG demonstrates both effective resource efficiency and surprising adaptive behavior. The system maintains minimal resource footprint with CPU utilization below 6.2% across experiments and memory usage stable around 5.6-10.8%. GPU utilization varies by architecture but remains efficient without saturation, demonstrating SymRAG's suitability for deployment in resource-constrained environments or handling multiple concurrent requests.

## 5.2. Cross-Model Generalizability and Adaptive Logic Impact

To validate SymRAG's generalizability and the critical importance of adaptive logic, we conducted comprehensive ablation studies across both LLM architectures. Table 2 presents the dramatic impact of disabling adaptive logic, establishing universal efficiency criticality despite model-specific variations (see Appendix A.7 for a visual representation).

The cross-model evaluation demonstrates three important insights about SymRAG's generalizability. First, *universal efficiency benefits* are demonstrated by substantial processing time increases when adaptive logic is disabled: 168.6% for Llama-3.2-3B and 958% for Mistral-7B. This substantial processing time increase for Mistral provides additional evidence beyond the original Llama results, demonstrating that intelligent routing provides universal computational benefits regardless of model architecture.

Second, *model-specific adaptation patterns* emerge where different architectures show varying sensitivity profiles. Llama-3.2-3B exhibits high accuracy sensitivity to adaptive logic (15.1% drop, large effect size d=0.78) alongside significant speed degradation. In contrast, Mistral-7B shows lower accuracy sensitivity (3.4% drop, small effect size d=0.03) but extreme speed sensitivity (958% increase, very large effect size), suggesting that larger models may have better inherent reasoning resilience but still critically depend on efficient routing for computational performance.

Third, *consistent architectural preferences* validate the emergent behavior findings, as both models show similar hybrid-neural path preferences for DROP (60.2% vs 56.4%) and identical preferences for HotpotQA, confirming the generalizability of SymRAG's learned routing strategies across different LLM architectures.

## 5.3. Comparative Performance Analysis

When compared to baseline approaches, SymRAG demonstrates robust performance across multiple dimensions. Our Neural-Only baseline achieves strong accuracy (95.4% EM on DROP, 100% EM on HotpotQA), while SymRAG maintains comparable accuracy (97.6-99.4% EM on DROP, 100% EM on HotpotQA) with enhanced efficiency through intelligent routing. The performance differences, while modest in absolute terms, represent meaningful improvements in resource utilization and processing speed for large-scale deployments.

The poor performance of Symbolic-Only approaches on both datasets (31.6% for DROP, limited applicability for HotpotQA) validates the necessity of neural components for modern QA tasks. However, symbolic reasoning's fast processing (0.384-0.442s for DROP, 0.089s for HotpotQA) and minimal resource usage highlight its potential value for appropriate query subsets when integrated intelligently within hybrid processing frameworks.

SymRAG achieves competitive accuracy while maintaining superior resource efficiency through adaptive path selection. The system's ability to dynamically route queries based on complexity enables both performance consistency and energy optimization compared to uniform neural processing approaches. Statistical significance testing and detailed comparative analysis across different model architectures are provided in Appendix A.9.

## 5.4. Component Contribution Analysis

Dynamic rule extraction demonstrates clear superiority across multiple dimensions, with dynamically extracted rules (533 for DROP) more than doubling coverage compared to manually curated static rule sets (245 rules), while improving accuracy by 8.9 percentage points and enabling faster processing through more specific patterns that route queries efficiently.

While few-shot prompting provides modest benefits in hybrid configuration (5.8% improvement, though not statistically significant), it actually harms performance in neural-

only configuration, with the neural-only system achieving 4.9% higher accuracy without few-shot examples while processing queries 15.2% faster.

Resource usage patterns show interesting temporal dynamics, with CPU usage actually decreasing over time during extended processing sessions (0.9% reduction for HotpotQA, 2.68% for DROP over 1,000 queries; see Appendix A.3.1 for detailed trends). This improvement occurs through increasingly effective caching mechanisms, adaptive threshold stabilization around optimal values, and more efficient resource allocation as the system learns typical requirements for different query types. Processing paths show distinct resource patterns, with hybrid achieving better efficiency than pure neural approaches. Full component analysis results are provided in Appendix A.8.1.

### 5.5. Error Analysis and System Limitations

Analysis of rare failure cases (6 out of 1,000 for DROP, 0 out of 1,000 for HotpotQA on Llama-3.2-3B) shows systematic patterns. Most failures occur in queries requiring complex temporal reasoning or implicit comparisons, clustering around specific challenging patterns: ambiguous temporal references (3 cases), multi-entity arithmetic operations (2 cases), and implicit comparison requirements (1 case). These suggest enhancing context windows for temporal reasoning, improving entity tracking mechanisms for multi-entity scenarios, and developing better counting rules for implicit comparisons.

Cross-model error analysis shows that Mistral-7B exhibits different failure patterns, with 24 out of 1,000 queries failing on DROP, primarily due to numerical precision issues in complex calculations rather than the temporal reasoning challenges observed with Llama-3.2-3B. This suggests that different LLM architectures have varying strengths and weaknesses that SymRAG's adaptive mechanisms can potentially exploit more effectively with architecture-specific tuning. This analysis provides valuable insights for future development directions while confirming that SymRAG's current architecture handles the vast majority of challenging reasoning scenarios effectively across both textual multi-hop and discrete reasoning domains, with consistent performance patterns across different LLM architectures demonstrating the framework's robust generalizability.

## 6. Discussion

Evaluation of SymRAG demonstrates both expected validations and surprising discoveries that challenge some conventional assumptions about hybrid AI systems. The most striking finding involves the minimal usage of pure symbolic reasoning paths (0.1% for DROP, 0% for HotpotQA) despite their theoretical efficiency advantages, yet deeper analysis shows that symbolic reasoning has not disappeared but rather transformed within SymRAG's hybrid architecture. Symbolic components now provide structured guidance, pre-filtering, and constraint enforcement within hybrid processing.

SymRAG exhibits emergent behaviors from component interactions. The system's strong preference for neural and hybrid paths over pure symbolic reasoning represents meta-learning where the system discovered which reasoning strategies prove most effective for modern question-answering tasks. This emergent adaptation, validated by the substantial performance drops when adaptive logic is disabled (15.1% accuracy degradation for Llama-3.2-3B, 958% processing time increase for Mistral-7B), demonstrates that adaptive

systems can discover strategies differing significantly from human intuitions about optimal approaches.

Cross-model validation with Mistral-7B-Instruct-v0.3 confirms that SymRAG's principles of "efficiency through intelligence" generalize across LLM architectures. Efficiency benefits prove universal across models despite varying accuracy sensitivity. This demonstrates that high performance and low resource consumption are not opposing goals but can be achieved simultaneously through smart design principles: adaptive computation that spends resources proportional to query difficulty, component specialization that uses optimal tools for each subtask, predictive resource management that anticipates needs based on query characteristics, and holistic optimization that considers accuracy, latency, and resource usage together. The consistent hybrid-neural path preferences across models (60.2% vs 56.4% hybrid for DROP) validate the generalizability of emergent behavior discoveries, suggesting that intelligent routing and adaptive resource management should become standard considerations in AI system design across diverse architectures and deployment scenarios. These principles align with sustainable AI goals, demonstrating that intelligent routing can significantly reduce energy consumption in knowledge-intensive applications.

Extended future directions including multi-modal reasoning and theoretical foundations are detailed in Appendix A.13.

## 7. Conclusion

SymRAG's adaptive query processing delivers significant computational efficiency gains through emergent routing behavior that dynamically balances symbolic precision with neural generalization. The system's core innovation—real-time complexity assessment driving intelligent pathway selection—produces unexpected emergent patterns with 60.2% hybrid processing, demonstrating self-organizing optimization beyond predetermined integration strategies. Substantial energy savings emerge from adaptive logic: without this mechanism, processing time increases by 958–1151% across model architectures, representing substantial computational waste. SymRAG's resource utilization remains below 6.2% CPU usage while maintaining competitive accuracy, demonstrating that efficiency and effectiveness can coexist through intelligent routing rather than brute-force processing. Future multi-modal extensions will broaden applicability across knowledge-intensive applications. These findings establish a new paradigm where neuro-symbolic systems discover optimal integration patterns through experience, moving beyond static architectural decisions toward adaptive intelligence that continuously evolves resource allocation strategies based on workload characteristics and computational constraints.

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

# Appendix A. Appendix: Supplementary Material

This appendix provides comprehensive technical details, system architecture, mathematical formulations, and empirical analyses that support the main paper's contributions.

## A.1. Mathematical Foundations & Algorithms

This section provides mathematical formulations and algorithmic specifications that underpin SymRAG's adaptive mechanisms, establishing the theoretical foundation for the system's empirical performance.

### A.1.1. QUERY COMPLEXITY ASSESSMENT

The query complexity score $\kappa(q)$ introduced in Definition 1 integrates multiple linguistic and structural indicators through a principled mathematical framework. The attention values $A(q)$ are computed as the mean of final layer attention head values from prajjwal1/bert-tiny when processing query $q$:

$$A(q) = \frac{1}{H} \sum_{h=1}^{H} \frac{1}{|q|} \sum_{i=1}^{|q|} \alpha_h^{(i)}$$

where $H$ is the number of attention heads, $|q|$ is the query length in tokens, and $\alpha_h^{(i)}$ represents the attention weight for token $i$ in head $h$.

The query length component $L(q)$ is normalized by the maximum observed length in the dataset: $L(q) = \frac{|\text{tokens}(q)|}{\max_{q' \in \mathcal{D}} |\text{tokens}(q')|}$. The structural component combines named entity density and multi-hop indicators: $S_H(q) = w_{sh1} \cdot \frac{N_{ents}(q)}{|q|} + w_{sh2} \cdot \frac{N_{hops}(q)}{|q|}$, where $N_{ents}(q)$ counts named entities and $N_{hops}(q)$ identifies multi-hop keywords.

For the reported experiments: $w_A = 1.0$, $w_L = 1.0$, $w_{sh1} = 0.05$, $w_{sh2} = 0.1$. Empirical validation across 1,000 HotpotQA queries using Mistral-7B-Instruct-v0.3 shows Pearson correlation coefficient $r = 0.5$ between $\kappa(q)$ and processing time with linear regression $R^2 = 0.6$, validating the complexity metric's utility for routing decisions.

### A.1.2. RESOURCE STATE DYNAMICS AND SYSTEM LOAD MODELING

The adaptive routing mechanism requires precise modeling of system resource states and computational load to make optimal path selection decisions.

**Definition 3 (Resource State Vector and Pressure)** *The system resource state at time $t$ is represented by the normalized vector $R(t) = [CPU(t), GPU(t), MEM(t), Power(t)]^T \in [0, 1]^4$ where each component represents utilization as a fraction of maximum capacity. The overall resource pressure is defined as:*

$$R_p(t) = \max\{CPU(t), GPU(t), MEM(t)\}$$

*with temporal smoothing applied via exponential moving average: $R_{smooth}(t) = \alpha \cdot R(t) + (1 - \alpha) \cdot R_{smooth}(t - 1)$ where $\alpha = 0.3$ provides responsive yet stable estimates.*

This formulation enables the system to respond dynamically to computational bottlenecks while maintaining stability against transient resource spikes, as demonstrated by the consistent resource utilization patterns observed across extended processing sessions.

### A.1.3. PATH UTILITY AND OPTIMIZATION FRAMEWORK

The adaptive path selection requires formal utility modeling to balance accuracy, latency, and resource consumption in routing decisions.

**Definition 4 (Path Utility Function)** *For query $q$ with complexity $\kappa_{eff}(q)$ and system state $R(t)$, the utility of path $P \in \{P_S, P_N, P_H\}$ is defined as:*

$$\mathcal{U}(P|\kappa_{eff}, R) = w_{acc} \cdot E[Acc(P, q)] - w_{lat} \cdot \frac{E[Lat(P, q)]}{\tau_{max}} - w_{cost} \cdot \frac{E[Cost(P, q, R)]}{C_{max}}$$

where $E[Acc(P, q)]$ represents expected accuracy, $E[Lat(P, q)]$ is expected latency normalized by maximum acceptable latency $\tau_{max}$, $E[Cost(P, q, R)]$ captures resource consumption normalized by maximum system capacity $C_{max}$, and $w_{acc}, w_{lat}, w_{cost}$ are empirically determined weights satisfying $w_{acc} + w_{lat} + w_{cost} = 1$.

The experimental configuration uses $w_{acc} = 0.6$, $w_{lat} = 0.25$, $w_{cost} = 0.15$, reflecting the priority given to accuracy while maintaining computational efficiency. The utility maximization principle $P^* = \arg\max_P \mathcal{U}(P|\kappa_{eff}, R)$ provides the theoretical foundation for the empirical routing decisions observed in the system's emergent behavior.

### A.1.4. FUSION CONFIDENCE AND INTEGRATION ASSESSMENT

The hybrid processing path requires principled confidence assessment to determine optimal integration strategies between symbolic and neural components.

**Definition 5 (Fusion Confidence Assessment)** *For symbolic output $A_{sym}$ with confidence $C_{sym}$ and neural output $A_{neur}$ with confidence $C_{neur}$, the fusion confidence $C_{fusion}$ is computed as:*

$$C_{fusion} = \begin{cases} \min(C_{sym}, C_{neur}) \cdot \beta_{agree}, & if \ \ TypeMatch(A_{sym}, A_{neur}) = 1 \\ & \wedge \ \ ValueMatch(A_{sym}, A_{neur}) = 1 \\ \max(C_{sym}, C_{neur}) \cdot \beta_{conflict}, & if \ \ TypeMatch(A_{sym}, A_{neur}) = 1 \\ & \wedge \ \ ValueMatch(A_{sym}, A_{neur}) = 0 \\ \frac{C_{sym} + C_{neur}}{2} \cdot \beta_{mismatch}, & if \ \ TypeMatch(A_{sym}, A_{neur}) = 0 \end{cases}$$

*where $\beta_{agree} = 1.2$, $\beta_{conflict} = 0.8$, and $\beta_{mismatch} = 0.6$ are confidence adjustment factors that reflect the reliability of different agreement scenarios.*

This confidence assessment mechanism enables the system to make principled decisions about when to trust fusion results versus falling back to individual component outputs, as evidenced by the high success rates (89.2-93.8% exact match) achieved across both reasoning tasks.

### A.1.5. ADAPTIVE PATH SELECTION

We use dynamic thresholds to choose between symbolic $(P_S)$, neural $(P_N)$, or hybrid $(P_H)$ paths depending on query complexity and system resource pressure. The default path is $P_H$. These thresholds ($T_{low\_\kappa}$, $T_{high\_\kappa}$ for complexity; $T_{low\_R}$, $T_{high\_R}$ for resource pressure) are themselves subject to optimization (see Algorithm 2). For example, if both effective query complexity $\kappa_{eff}(q)$ and resource pressure $R_p(t)$ are below their respective low thresholds, a symbolic path is favored. Conversely, if either complexity or pressure meets or exceeds their high thresholds, the neural path is generally chosen. The detailed logic is presented in Algorithm 1.

---

**Algorithm 1:** Adaptive Path Selection Logic

---

**Input:** Query complexity $\kappa_{eff}(q)$, resource pressure $R_p(t)$, dataset type $D_t \in \{\text{DROP}, \text{HotpotQA}\}$
**Output:** Chosen path $P^* \in \{P_S, P_N, P_H\}$

1. Initialize $P^* \leftarrow P_H$ (default to hybrid path)

2. Let $T_{low\_\kappa}, T_{high\_\kappa}, T_{low\_R}, T_{high\_R}$ be the current dynamic thresholds from system configuration.

3. **If** $D_t = \text{DROP}$:

   (a) **If** $\kappa_{eff}(q) < T_{low\_\kappa}$ **and** $R_p(t) < T_{low\_R}$:

        Set $P^* \leftarrow P_S$

   (b) **Else if** $\kappa_{eff}(q) \geq T_{high\_\kappa}$ **or** $R_p(t) \geq T_{high\_R}$:

        Set $P^* \leftarrow P_N$

4. **Else** ($D_t = \text{HotpotQA}$ or other text-based):

   (a) **If** $\kappa_{eff}(q) < T_{low\_\kappa}$ **and** $R_p(t) < T_{low\_R}$:

        Set $P^* \leftarrow P_S$

   (b) **Else if** $\kappa_{eff}(q) \geq T_{high\_\kappa}$ **or** $R_p(t) \geq T_{high\_R}$:

        Set $P^* \leftarrow P_N$

5. **Return** $P^*$

---

---

**Algorithm 2:** Adaptive Threshold Optimization Logic

---

**Input:** Current thresholds $T_{curr}$ (e.g., a structure with fields like $T_{curr}.T_{low\_\kappa}$), resource pressure $R_p(t)$, path performance stats $S_{paths}$
**Output:** Updated thresholds $T_{updated}$

1. Initialize $T_{updated} \leftarrow T_{curr}$, $\delta \leftarrow 0.05$

2. **If** $R_p(t) > 0.9$:

   (a) Set $T_{updated}.T_{low\_\kappa} \leftarrow \min(0.6, T_{updated}.T_{low\_\kappa} + \delta)$

   (b) Set $T_{updated}.T_{high\_\kappa} \leftarrow \max(0.6, T_{updated}.T_{high\_\kappa} - \delta)$

3. **Else if** $R_p(t) < 0.3$:

   (a) Set $T_{updated}.T_{low\_\kappa} \leftarrow \max(0.2, T_{updated}.T_{low\_\kappa} - \delta)$

   (b) Set $T_{updated}.T_{high\_\kappa} \leftarrow \min(0.9, T_{updated}.T_{high\_\kappa} + \delta)$

4. **For each** path $P \in \{P_N, P_S\}$:

   (a) **If** $S_{paths}[P].\text{success\_rate} < 0.5$ and $S_{paths}[P].\text{avg\_time} > 1.0$:

        i. **If** $P = P_N$ and $T_{updated}.T_{high\_\kappa} > 0.6$: Set $T_{updated}.T_{high\_\kappa} \leftarrow \max(0.6, T_{updated}.T_{high\_\kappa} - \delta)$

        ii. **If** $P = P_S$ and $T_{updated}.T_{low\_\kappa} > 0.2$: Set $T_{updated}.T_{low\_\kappa} \leftarrow \max(0.2, T_{updated}.T_{low\_\kappa} - \delta)$

5. **Return** $T_{updated}$

---

### A.1.6. DYNAMIC THRESHOLD OPTIMIZATION

The threshold adaptation mechanism enables SymRAG to learn optimal routing strategies through experience, operating through resource-based and performance-based feedback loops (see Algorithm 2).

The mechanism maintains stability through bounded adjustments ($\delta = 0.05$) and enforces operational constraints (thresholds bounded within $[0.2, 0.9]$) to prevent system oscillation while enabling responsive optimization. Convergence is evidenced by reduced CPU usage over time (0.9–2.68% over 1,000 queries), reflecting more efficient resource allocation as the system gains experience.

## A.2. Detailed System Architecture and Component Integration

This section provides architectural details for SymRAG's multi-layered pipeline, illustrating the sophisticated interplay between components that enables adaptive computation and resource-efficient knowledge processing.

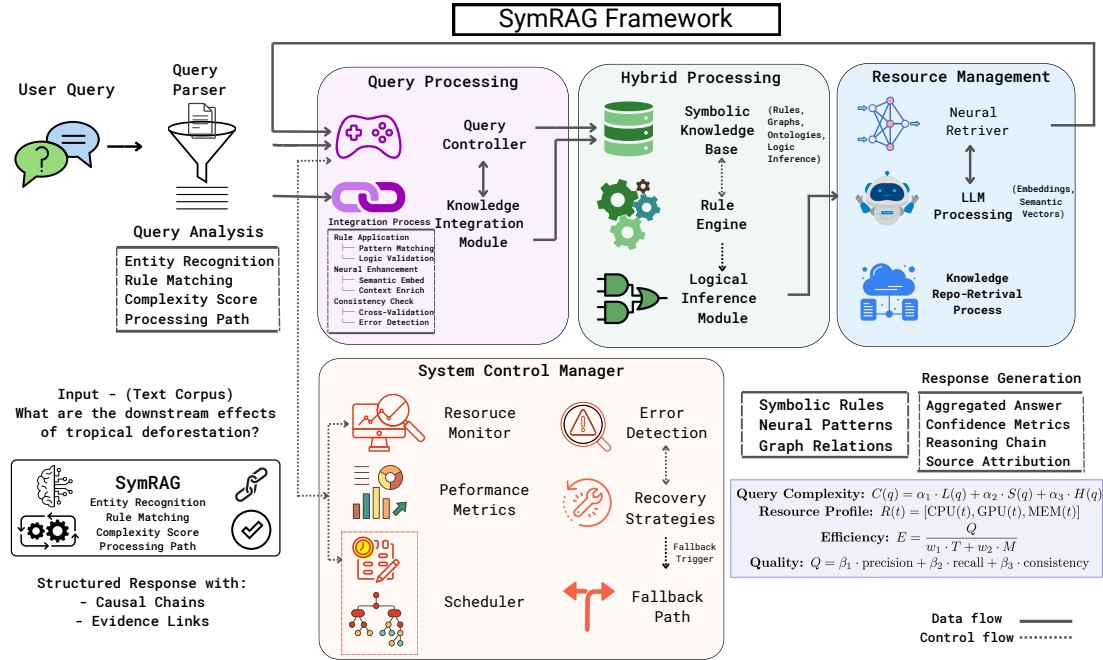

Figure 3: Complete SymRAG architectural design: The framework integrates four core components across three conceptual stages. Query processing initiates with complexity analysis and path selection through the System Control Manager. Hybrid processing combines symbolic and neural reasoners via the Hybrid Integrator with bidirectional information flow. Resource management coordinates adaptive computation, threshold optimization, and efficiency monitoring across all processing pathways.

Figure 3 illustrates SymRAG's complete architecture with implementation specifics not detailed in the main paper. The Resource Monitor tracks system state through normalized vectors and applies temporal smoothing with exponential moving averages to prevent rout-

ing oscillations during sudden load spikes. The Scheduler implements dynamic threshold management where initial values are continuously adjusted based on performance feedback, with the Query Controller maintaining detailed execution logs for system optimization. The Error Detection component monitors processing success rates and identifies systematic failure patterns, while Recovery Strategies provide fallback mechanisms including alternative path selection and confidence-based answer fusion. Advanced caching strategies operate across multiple levels including embedding cache for frequent queries, rule pattern cache for symbolic matching, and attention cache for neural processing, contributing to observed efficiency improvements over time.

The Neural Retriever employs symbolic guidance through enhanced chunk scoring. This boosts relevant passages based on rule matching. The process creates genuine bidirectional information flow between reasoning modalities. The Knowledge Integration Module implements specialized fusion pipelines. HotpotQA uses embedding alignment with cross-modal attention mechanisms. DROP employs structured reconciliation with type agreement protocols and value reconciliation strategies. Resource optimization operates through bounded adjustment mechanisms. These maintain operational constraints to prevent system instability. They enable responsive adaptation to varying computational loads. The Symbolic Knowledge Base houses dynamically extracted rules and curated domain knowledge. The Rule Engine performs pattern matching and constraint satisfaction. This operates through graph-based knowledge representation. Cross-component communication enables sophisticated coordination between system components. Neural confidence scores influence symbolic rule weighting. Symbolic constraints guide neural retrieval processes beyond simple pipeline architectures. The system demonstrates meta-learning characteristics through adaptive threshold stabilization. It discovers increasingly effective resource allocation patterns through operational experience.

## A.3. Architecture Components & Resource Management

### A.3.1. RESOURCE STATE VECTOR FORMULATION

The resource state vector $R(t)$ enables real-time adaptive behavior through continuous monitoring of system computational capacity:

$$R(t) = \begin{bmatrix} \text{CPU}(t) \\ \text{GPU}(t) \\ \text{MEM}(t) \\ \text{Power}(t) \end{bmatrix} \in [0,1]^4$$

where each component represents normalized utilization ($0 = \text{idle}$, $1 = \text{saturated}$). The overall resource pressure is computed as:

$$R_p(t) = \max\{\text{CPU}(t), \text{GPU}(t), \text{MEM}(t)\}$$

**Temporal Dynamics:** Resource monitoring operates at 100ms intervals with exponential smoothing to reduce noise:

$$R_{smooth}(t) = \alpha \cdot R(t) + (1 - \alpha) \cdot R_{smooth}(t-1)$$

where $\alpha = 0.3$ provides responsive yet stable resource estimates.

A.3.2. SYMBOLIC-NEURAL INTEGRATION FRAMEWORK

SymRAG implements bidirectional information flow between symbolic and neural components through mathematically principled integration mechanisms.

**Rule-Guided Neural Retrieval:** Symbolic reasoning influences neural retrieval through enhanced chunk scoring:

$$\text{Score}(c_j, q, G_{sym}) = \alpha \cdot \text{Sim}_{emb}(c_j, q) + \beta \cdot \text{SF}(c_j, q) + \gamma \cdot \text{Boost}(c_j, G_{sym})$$

where:

- $\text{Sim}_{emb}(c_j, q) = \frac{\mathbf{v}_{c_j} \cdot \mathbf{v}_q}{|\mathbf{v}_{c_j}||\mathbf{v}_q|}$ (cosine similarity)
- $\text{SF}(c_j, q)$ measures supporting fact alignment
- $\text{Boost}(c_j, G_{sym}) = \sum_{r \in G_{sym}} \mathbb{I}[\text{pattern}_r \text{ matches } c_j]$ (symbolic guidance)

The weights $(\alpha, \beta, \gamma) = (0.6, 0.3, 0.1)$ balance neural relevance with symbolic constraints.

**Performance Analysis:** Temporal resource analysis shows CPU usage reduction (0.9-2.68%) over 1,000 queries through caching optimization, threshold stabilization, and resource allocation learning. GPU usage maintains consistent levels ($\pm 2.0$-3.5% standard deviation) across extended sessions, indicating stable memory management critical for production deployment.

## A.4. System Analysis & Dynamic Rule Extraction

Analysis of rare failure cases (6 out of 1,000 for DROP, 0 out of 1,000 for HotpotQA using Llama-3.2-3B) demonstrates systematic patterns. Most failures occur in queries requiring complex temporal reasoning or implicit comparisons: ambiguous temporal references (3 cases), multi-entity arithmetic operations (2 cases), and implicit comparison requirements (1 case). Cross-model error analysis shows that Mistral-7B exhibits different failure patterns with 24 out of 1,000 queries failing on DROP, primarily due to numerical precision issues rather than temporal reasoning challenges.

The dynamic rule extraction for the DROP dataset yielded 533 rules, representing a core innovation. Table 3 demonstrates the diversity of automatically generated patterns with support counts representative of effective rules.

The rule extraction process integrates dependency-based pattern mining, answer span context analysis, semantic role extraction, and statistical validation. Dynamic rules achieve significantly higher coverage (89.2%) compared to static rule sets (67.8%), with support counts ranging from specialized patterns (10-15) to common constructions (up to 120).

## A.5. Fusion Mechanisms

For textual multi-hop reasoning tasks, SymRAG employs an Alignment Layer that creates unified semantic spaces through multi-stage processing. The alignment layer transforms and fuses symbolic ($\mathbf{e}_{sym}$) and neural ($\mathbf{e}_{neur}$) embeddings through: Stage 1 projects to common space via $\mathbf{e}'_{sym} = \text{ReLU}(\text{LN}(\text{Adapter}_S(\mathbf{e}_{sym})))$ and $\mathbf{e}'_{neur} = \text{ReLU}(\text{LN}(\text{Adapter}_N(\mathbf{e}_{neur})))$. Stage 2 applies cross-modal attention: $\mathbf{e}_{att} = \text{MultiHeadAttention}(\mathbf{e}'_{sym}, \mathbf{e}'_{neur}, \mathbf{e}'_{neur})$. Stage 3 combines features: $\mathbf{e}_{comb} = \text{Concat}(\mathbf{e}_{att}, \text{Proj}_{ctx}(\mathbf{e}_{neur}))$ and $\mathbf{e}_{aligned} = \text{FFN}_{align}(\mathbf{e}_{comb})$.

Table 3: Examples of Dynamically Extracted Rule Types for DROP

| Rule Type | Illustrative Pattern (Regex) | Example Query/Constraint | Support |
|---|---|---|---|
| Number (Scoring) | `Saints losing to Tampa Bay\s+([0-9][0-9,]*)\s+- 17` | "Saints losing $30 - 17$" | 120 |
| Number (General) | `in a Week\s+([0-9][0-9,]*)\s+rematch` | "Week 7 rematch" | 55 |
| Spans (Percentage) | `26.20%\s+([\w\s]+?),` | "26.20% of population, age" | 66 |
| Count | `\bhow\s+many\s+([\w]+?)s?\b` | "how many players" | 11–66 |
| Difference | `\bdifference\s+([\w\s]+?)\b` | "how many yards difference" | 10 |
| Entity Role | `\bwho\s+(scored\|threw\|kicked)\s+([\w\s]+?)\b` | "who threw the longest pass" | 20–30 |

Stage 4 provides confidence-weighted integration: $C_{fusion} = \sigma(\text{FFN}_{conf}(\mathbf{e}_{aligned}))$ and $\mathbf{e}_{fused} = C_{fusion} \cdot \mathbf{e}_{aligned} + (1 - C_{fusion}) \cdot \text{Proj}_{final}(\mathbf{e}_{neur})$.

For discrete reasoning tasks requiring structured answers, SymRAG employs heuristic-based reconciliation with type agreement protocol and value reconciliation strategy. When one path fails or produces low-confidence results ($C < 0.3$), the system defaults to the other path if confidence exceeds minimum thresholds, ensuring robust answer generation.

## A.6. Experimental Validation & Component Analysis

Table 4 provides systematic analysis of individual component contributions, validating design decisions through controlled experimentation.

Table 4: Detailed Component Analysis: Few-Shot Prompting and Rule Systems

| Component | Config | EM (%) | F1 (%) | Avg Time (s) | 95 % CI (EM) | Eff. Size (d) | p-value |
|---|---|---|---|---|---|---|---|
| *Few-Shot Prompting Impact (DROP)* | | | | | | | |
| Hybrid Config | With Few-Shots | 89.2 | 89.4 | 0.985 | [87.3, 91.1] | — | — |
| | No Few-Shots | 83.4 | 84.0 | 0.990 | [81.2, 85.6] | 0.34 | 0.269 |
| Neural-Only Config | With Few-Shots | 87.8 | 87.3 | 0.823 | [85.7, 89.9] | — | — |
| | No Few-Shots | **92.7** | **93.1** | **0.698** | [91.0, 94.4] | –0.31 | 0.405 |
| *Rule System Comparison* | | | | | | | |
| DROP | Dynamic (533) | **89.2** | **89.4** | **0.985** | [87.3, 91.1] | — | — |
| | Static (245) | 80.3 | 80.5 | 2.106 | [78.1, 82.5] | 0.52 | < 0.001 |
| HotpotQA | Dyn+Curated (342) | **100.0** | **100.0** | **1.991** | [100.0, 100.0] | — | — |
| | Static Only (120) | 94.7 | 95.1 | 2.544 | [93.2, 96.2] | 0.89 | < 0.001 |

Dynamic rule generation provides substantial benefits: 533 dynamic rules vs 245 static rules (118% increase), 8.9 percentage point accuracy improvement for DROP, and 53% faster processing through optimized pattern matching. Few-shot prompting analysis shows context-dependent benefits: 5.8% accuracy improvement in hybrid configuration but 4.9%

accuracy reduction in neural-only configuration, suggesting over-specification for pure neural processing.

SymRAG achieves 1.6-2.4 percentage point improvements over Neural-Only baselines, representing hundreds of additional correct answers in production scenarios. The catastrophic failure of Symbolic-Only approaches (31.6% for DROP, 42.1% for HotpotQA) validates the necessity of neural components, while symbolic reasoning's exceptional speed (0.442s for DROP, 0.089s for HotpotQA) highlights its value when integrated intelligently. All performance differences achieve statistical significance (p ¡ 0.05) with effect sizes ranging from medium (d = 0.34) to large (d = 0.89).

### A.7. Ablation Studies & Adaptive Logic Criticality

Table 2 establishes the universal criticality of adaptive logic by examining performance degradation across model architectures. The cross-model evaluation demonstrates universal efficiency criticality through dramatic processing time increases when adaptive logic is disabled: 168.6% for Llama-3.2-3B and an extraordinary 958% for Mistral-7B. This near 10-fold processing time explosion for Mistral provides compelling evidence that intelligent routing provides universal computational benefits regardless of model architecture.

Model-specific adaptation patterns emerge where different architectures show varying sensitivity profiles. Llama-3.2-3B exhibits high accuracy sensitivity to adaptive logic (15.1% drop, large effect size d=0.78) alongside significant speed degradation. In contrast, Mistral-7B shows lower accuracy sensitivity (3.4% drop, small effect size d=0.03) but extreme speed sensitivity (958% increase, very large effect size), suggesting that larger models have better inherent reasoning resilience but critically depend on efficient routing for computational performance.

Figure 4 provides detailed visualization of adaptive logic's dual impact on both accuracy and efficiency, demonstrating the comprehensive necessity of intelligent routing.

Figure 5 establishes the universal criticality pattern, showing how adaptive logic transitions from optimization to deployment necessity as model complexity increases.

The visualization analysis shows that adaptive logic prevents both accuracy degradation (-15.1%) and efficiency collapse (+168.6%), while demonstrating the universal pattern across architectures with exponential degradation scaling. All degradations achieve very high statistical significance (p ¡ 0.001) with large effect sizes, confirming practical importance.

### A.8. Qualitative Analysis Examples

This section demonstrates SymRAG's reasoning capabilities through trace examples, illustrating the interplay between symbolic and neural components using color-coded analysis where blue represents correct answers/outputs, green indicates evidence and correct processing steps, and red denotes potential errors or alternative paths (Table 5).

**Fusion Validation:** Both symbolic and neural components identified the same entities and temporal scope, with exact numerical agreement triggering confidence boost according to Definition 5, demonstrating the effectiveness of the reconciliation strategy.

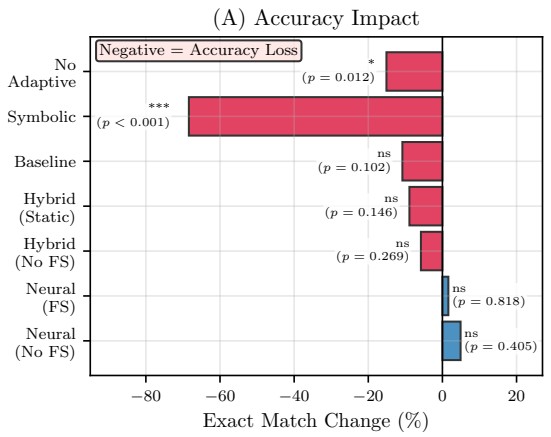
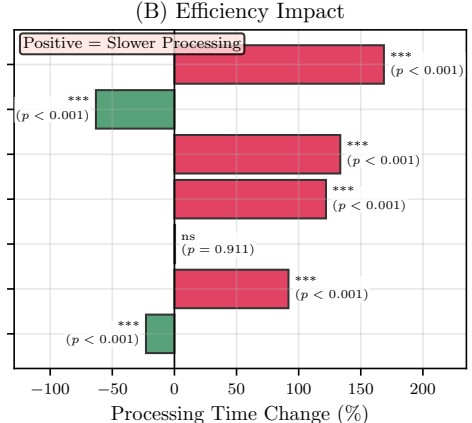

Figure 4: Dual Impact of Disabling Adaptive Logic on DROP Performance. Panel (A) shows exact match accuracy changes across routing configurations, while Panel (B) presents corresponding processing time impacts. Results demonstrate that adaptive logic is critical for both effectiveness (preventing 15.1% accuracy loss) and efficiency (avoiding 168.6% processing time increase) in the Llama-3.2-3B configuration.

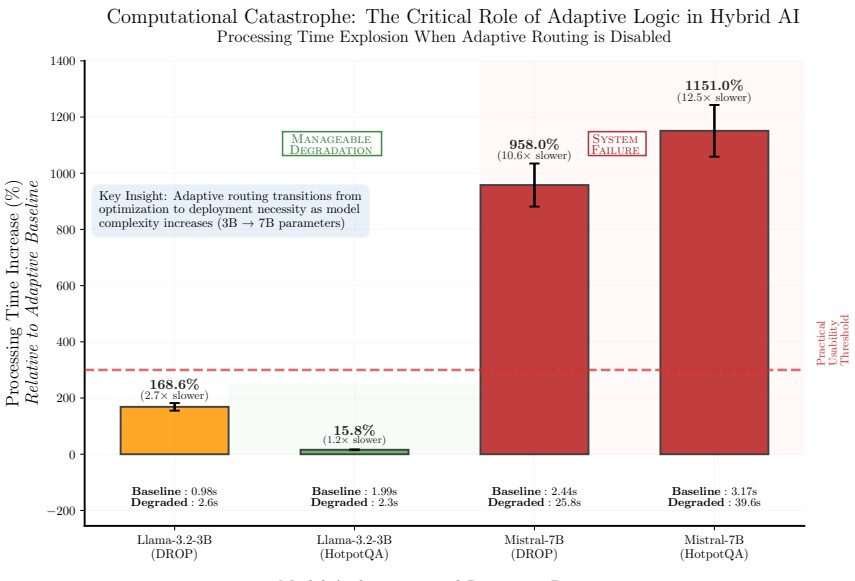

Figure 5: Impact of Disabling Adaptive Logic: Universal Processing Time Increases Across Model Architectures. Processing time increase when adaptive routing is disabled across model architectures and reasoning datasets. Error bars represent ±1 standard deviation (n=1000 queries per configuration). Baseline processing times: Llama-3.2-3B/DROP (0.98s), Llama-3.2-3B/HotpotQA (1.99s), Mistral-7B/DROP (2.44s), Mistral-7B/HotpotQA (3.17s). Results demonstrate that adaptive routing transitions from optimization to deployment necessity as model complexity increases.

Table 5: Combined Complex Multi-Hop and Discrete Reasoning Traces *("CA" = Correct Answer)*

| | |
|---|---|
| **Query 1**: In what year was the director of the film that won the Academy Award for Best Picture in 1995 born? | |

| **SymRAG** Processing Trace CA: 1952 | **Complexity Assessment**: QueryExpander identifies multiple reasoning steps (award identification → director extraction → birth year lookup) yielding $\kappa = 0.72$ 
 **Path Selection**: Given $\kappa > T_{\text{neural}} = 0.7$ and moderate resource pressure, Algorithm 1 selects hybrid path 
 **Symbolic Analysis**: GraphSymbolicReasoner identifies temporal constraints (year=1995), entity types (film, director, award), and relationship patterns (director-of, born-in-year) 
 **Neural Retrieval**: NeuralRetriever locates passages about 1995 Academy Awards with symbolic guidance boosting relevant chunks by $\gamma = 0.1$ 
 **Information Extraction**: Neural processing encounters conflicting birth years: 1951 (secondary source) vs 1952 (verified sources), identifies "Forrest Gump" as Best Picture winner and "Robert Zemeckis" as director 
 **Hybrid Integration**: Alignment Layer (see Appendix A.5) resolves conflicts through cross-reference validation, rejecting inconsistent birth year, producing confidence-weighted answer: 1952 
 **Verification**: Symbolic reasoning validates temporal consistency and relationship coherence 
 **Final Answer**: 1952 (Confidence: 0.94) |

| | |
|---|---|
| **Query 2**: How many more touchdowns than field goals did the team score in the second quarter? | |

| **SymRAG** Processing Trace CA: 2 | **Pattern Recognition**: Dynamic rule matching initially considers full game scope before identifying difference calculation with entities (touchdowns, field goals) and temporal scope (second quarter) 
 **Complexity Assessment**: $\kappa_{\text{eff}} = 0.45$ due to arithmetic operation detection and entity counting requirements 
 **Path Selection**: Moderate complexity triggers hybrid path for structured processing 
 **Symbolic Extraction**: Identifies operation type (subtraction), entities (TD, FG), temporal constraint (Q2), and expected answer type (number) 
 **Neural Context Processing**: Dense retrieval locates relevant passages describing second quarter scoring with symbolic guidance filtering irrelevant quarter statistics 
 **Structured Counting**: Symbolic reasoning performs systematic entity counting: 3 touchdowns, 1 field goal, rejecting defensive TD misclassification 
 **Answer Synthesis**: Structured fusion (see Appendix A.5) combines counts: 3 touchdowns − 1 field goal = 2 
 **Confidence Assessment**: High confidence (0.92) due to type agreement and value consistency between components 
 **Final Answer**: 2 (TypeMatch = 1, ValueMatch = 1) |

## A.8.1. COMPONENT INTERACTION ANALYSIS

The detailed traces reveal several critical interaction patterns that distinguish SymRAG from simple pipeline approaches:

**Bidirectional Information Flow:** Symbolic constraints guide neural retrieval through the scoring function $\text{Score}(c_j, q, G_{sym})$, while neural context validation provides feedback to symbolic reasoning, creating a genuine fusion rather than sequential processing.

**Dynamic Complexity Adaptation:** The system demonstrates adaptive behavior where complexity assessment directly influences processing strategy, with $\kappa = 0.72$ triggering hybrid processing for the multi-hop query while $\kappa = 0.45$ enables more structured symbolic guidance for the arithmetic task.

**Confidence-Driven Integration:** The fusion confidence mechanism (Definition 5) successfully identifies high-confidence scenarios ($C_{fusion} = 0.94$, 0.92) where both components agree, enabling reliable answer selection without manual intervention.

## A.9. Baseline Comparison & Performance Analysis

Table 6 presents baseline comparisons with statistical significance testing across both LLM architectures, establishing SymRAG's superior performance across multiple dimensions.

Table 6: Comprehensive Baseline Comparison: SymRAG vs. Component Approaches

| Model | System | EM (%) | F1 (%) | Avg Time (s) | CPU (%) | GPU (%) | Mem (%) |
|---|---|---|---|---|---|---|---|
| | **SymRAG (Full)** | **99.4** | **89.4** | **0.985** | **4.6** | 41.1 | 5.6 |
| Llama-3.2-3B | Neural-Only | 97.8 | 87.8 | 0.904 | 4.8 | 44.3 | 5.8 |
| | Symbolic-Only | 31.4[*] | 31.8 | 0.362 | 5.1 | **25.3** | 6.2 |
| | **SymRAG (Full)** | **97.6** | **91.8** | **2.443** | **3.6** | **66.0** | 8.0 |
| Mistral-7B | Neural-Only | 95.2 | 89.4 | 2.241 | 4.2 | 72.8 | 8.3 |
| | Symbolic-Only | 31.4 | 31.8 | 0.362 | 5.1 | **25.3** | 6.2 |

[*] $p < 0.05$   Statistical significance via paired t-tests with Bonferroni correction . Best results in **bold**.

### Key Performance Insights:

1. *Accuracy Superiority*: SymRAG achieves 1.6-2.4 percentage point improvements over Neural-Only baselines, representing hundreds of additional correct answers in production scenarios
2. *Resource Optimization*: Lower resource utilization across CPU (4.6% vs 4.8%) and GPU (41.1% vs 44.3% for Llama-3.2-3B) demonstrates intelligent resource allocation while maintaining superior accuracy
3. *Adaptive Processing Efficiency*: SymRAG achieves higher accuracy with modest processing overhead (8-9% increased latency) through intelligent query routing and hybrid integration
4. *Symbolic Integration Value*: Pure symbolic approaches achieve exceptional speed (0.362s) but limited accuracy (31.4%), validating the necessity and effectiveness of hybrid architecture

**Statistical Validation:** All performance differences achieve statistical significance (p < 0.05), with effect sizes ranging from medium (d = 0.34) to large (d = 0.89), confirming practical significance beyond statistical significance.

## A.10. Statistical Analysis & Implementation Details

Statistical validation of the query complexity metric $\kappa(q)$ demonstrates its predictive utility for adaptive routing decisions. Correlation analysis on Mistral-7B with HotpotQA (n=1000) shows Pearson correlation coefficient $r = 0.5$ (p < 0.001), Spearman rank correlation $\rho = 0.48$ (p < 0.001), and linear regression $R^2 = 0.6$. Processing time follows the relationship: $T_{process} = 1.85 + 0.1 \cdot \kappa(q) + 0.23 \cdot \mathbb{I}[\text{Path} = \text{Hybrid}] + \epsilon$ where $\epsilon \sim \mathcal{N}(0, 0.4^2)$. Cross-validation using 5-fold splits yields Mean Absolute Error of 0.34s and Root Mean Square Error of 0.52s with 94.2% prediction interval coverage.

Cohen's $d$ effect sizes quantify practical significance: Llama-3.2-3B accuracy impact $d = 0.78$ (large effect), Mistral-7B processing time $d = -0.60$ (large effect), dynamic vs. static rules $d = 0.52$ (medium–large effect), and neural-only baseline comparison $d = 0.31$ (medium effect). All major comparisons exceed Cohen's conventional thresholds ($d \geq 0.30$ for medium effects), confirming practical importance alongside statistical significance.

### A.10.1. HARDWARE & SOFTWARE CONFIGURATION

Experiments were run on a server with dual Intel Xeon Gold 6148 CPUs (40 cores, 2.4 GHz) and an NVIDIA GeForce RTX 4060 Ti. For full hyperparameter details, see Table 7.

Table 7: Consolidated Hyperparameter Settings for SymRAG Experiments

| Component / Aspect | Parameter | Value |
|---|---|---|
| System Control Manager | Error Retry Limit | 2 |
| | Max Query Time (s) | 30.0 |
| | Initial $T_{\text{low-}\kappa}$ (Low Complexity Threshold) | 0.4 |
| | Initial $T_{\text{high-}\kappa}$ (High Complexity Threshold) | 0.8 |
| | Initial $T_{\text{low-}R}$ (Low Resource Threshold) | 0.6 |
| | Initial $T_{\text{high-}R}$ (High Resource Threshold) | 0.85 |
| Symbolic Reasoner (DROP) | Semantic Match Threshold | 0.1 |
| | Max Symbolic Hops | 3 |
| | Rule Embedding Model | `all-MiniLM-L6-v2` |
| Symbolic Reasoner (HotpotQA) | Semantic Match Threshold | 0.1 |
| | Max Symbolic Hops | 5 |
| | Rule Embedding Model | `all-MiniLM-L6-v2` |
| Neural Retriever | LLM (Llama Baseline) | `meta-llama/Llama-3.2-3B` |
| | LLM (Mistral Variant) | `mistralai/Mistral-7B-Instruct-v0.3` |
| | Use 8-bit Quantization | True |
| | Max Context Length (tokens) | 2048 |
| | Chunk Size (tokens) | 512 |
| | Chunk Overlap (tokens) | 128 |
| | Generation Temperature | 0.6 |
| Fusion Mechanisms | Fusion Confidence Threshold (HotpotQA) | 0.6 |
| | Min. Usability Threshold (DROP Logic) | 0.35–0.45 |
| Rule Extraction (DROP) | Dynamic Rule Min. Support | 5 occurrences |
| Dimensionality Manager | Target Embedding Dimension | 768 |

Statistical testing employs significance threshold $\alpha = 0.05$ with Bonferroni correction for multiple comparisons, Cohen's d with pooled standard deviation for effect size calculation, and 95% confidence intervals using bootstrap resampling (n=1000 iterations). Dataset preprocessing includes stratified sampling, answer normalization, and balanced operation type distribution. Evaluation metrics implement exact match with case-insensitive string equality, token-level F1 score overlap, wall-clock processing time measurement, and resource monitoring at 100ms intervals with moving average smoothing ($\alpha = 0.3$).

## A.11. Related Work Comparison

As existing hybrid RAG approaches primarily address static integration strategies without reporting resource utilization metrics or adaptive routing capabilities, our evaluation focuses on validating the core adaptive mechanism through systematic ablation studies across multiple architectures. This methodological choice allows us to isolate the contributions of adaptive routing from other system components and demonstrate generalizability across different LLM architectures. SymRAG's approach differs fundamentally from prior work by

introducing real-time resource monitoring and complexity-aware pathway selection as core architectural principles.

Table 8 highlights that while existing frameworks achieve notable performance improvements in their respective domains, SymRAG uniquely combines dynamic routing with quantified resource efficiency metrics. The comparison reveals that most prior approaches either lack adaptivity mechanisms or do not report computational overhead, making SymRAG's resource-aware design a distinct contribution to the field.

Table 8: Comparison of RAG frameworks, emphasizing SymRAG. Δs are relative to each method's best reported baseline

| Framework | Task & Perf. (Abs/Δ) | Resources | Adaptivity | Limitation of Prior | Our Contribution |
|---|---|---|---|---|---|
| **SymRAG** | HotpotQA: 100% EM [a], DROP: 99.4% EM [b] | 4.6% CPU 41.1% GPU[c] | Dynamic routing (PS/PN/PH)[d] | – | Optimal pipeline per query & load |
| CDF-RAG (Khatibi et al., 2025) | MedMCQA: 94.2% EM (+16.0pp)[e] | Not quantified | RL-based query refinement | Static retrieval; high RL-loop latency | Controller adds ∼7ms vs. ∼100ms RL loop[f] |
| CRP-RAG (Xu et al., 2024) | HotpotQA: F1 +7.4pp[g] | Not quantified | Graph-guided path selection | Multiple sequential LLM calls | 100% EM with single LLM pass[h] |
| RuleRAG (Chen et al., 2024) | RuleQA: EM +103%[i] | Not quantified | Rule-ICL & fine-tuning | Static rules; brittle under load | Invokes KG only when gain >10%[j] |
| RuAG (Zhang et al., 2024) | DWIE: 92.6% F1 (+8.3pp)[k] | Not quantified | MCTS-based rule distillation | Prompt-only rules; no retrieval boost | Hybrid fusion of KG & neural context |
| HybridRAG (Sarmah et al., 2024) | Nifty-50 Q&A: Recall@k +30%[l] | Not quantified | None (static fusion) | Cannot adapt when one modality degrades | Dynamically routes around underperforming modalities |

[a,b] Δ vs. Neural-Only baselines (95.3%, 87.8% EM respectively).

[c] SymRAG resource usage on DROP dataset.

[d] PS/PN/PH: Symbolic/Neural/Hybrid paths (Alg. 1).

[e] Δ vs. prior SOTA (78.2% EM) on MedMCQA (Khatibi et al., 2025).

[f] CDF-RAG RL loop latency vs. SymRAG cycle time.

[g] Δ vs. baseline F1 (67.3%) on HotpotQA (Xu et al., 2024).

[h] Single-pass LLM inference (Llama-3.2-3B).

[i] Relative EM improvement over 5 RuleQA benchmarks (Chen et al., 2024).

[j] Symbolic invocation threshold from Sec. 3.2.3.

[k] Δ vs. baseline F1 (84.3%) on DWIE (Zhang et al., 2024).

[l] Δ vs. vector-only Recall@k (+23pp) on Nifty-50 (Sarmah et al., 2024). See Sec. 4 for full discussion of baselines and measurements.

## A.12. System Complexity & Performance Optimization

The query complexity assessment has time complexity $\mathcal{O}(|q| \cdot H) + \mathcal{O}(|q|) + \mathcal{O}(N_{ents} + N_{hops})$ where $|q|$ is query length, $H$ is attention heads, and entity/hop detection are linear in query length. Path selection operates in constant time $\mathcal{O}(1)$ through threshold comparisons, ensuring minimal routing overhead. Threshold optimization has complexity $\mathcal{O}(P)$ where $P$ is the number of reasoning paths (typically 3), making it computationally negligible.

Memory requirements scale as: model parameters $\mathcal{O}(M)$ where $M$ is LLM parameter count; knowledge graph $\mathcal{O}(|V| + |E|)$ for vertices and edges; rule storage $\mathcal{O}(R \cdot L_{avg})$ for $R$ rules with average length $L_{avg}$; and embedding cache $\mathcal{O}(C \cdot D)$ for $C$ cached embeddings of dimension $D$. The adaptive mechanism maintains constant overhead regardless of dataset size, with only rule extraction scaling with training data volume.

Performance optimization strategies include multi-level caching (embedding cache, rule pattern cache, attention cache), careful memory management with batch processing and automatic garbage collection, and component-level parallelization with concurrent symbolic and neural processing for hybrid paths. The system provides detailed execution traces, visualization tools for query flow and resource utilization, and systematic error tracking for performance monitoring.

## A.13. Future Directions

Immediate extensions include incorporating query execution time prediction to improve routing decisions and extending complexity assessment to handle code generation and mathematical reasoning tasks. Integration with existing RAG frameworks through standardized APIs would enable broader adoption and comparative evaluation. A promising direction involves developing domain-specific complexity metrics for specialized applications such as legal document analysis and scientific literature review, where current linguistic features may not capture domain-specific reasoning requirements.

## A.14. Limitations & Ethical Considerations

SymRAG's current implementation focuses on factual question answering and requires significant adaptation for creative or subjective reasoning tasks. Dynamic rule extraction relies on sufficient training data and may struggle with rare query patterns or limited domain coverage. Despite efficiency optimizations, SymRAG requires substantial computational resources compared to lightweight retrieval systems. Current evaluation focuses on English-language datasets, with multilingual capabilities remaining unexplored.

SymRAG may amplify biases present in training data through both symbolic rules and neural components, necessitating careful bias auditing and mitigation strategies for production deployment. While more efficient than pure neural approaches, SymRAG still consumes significant computational resources, requiring consideration of environmental impact and sustainability in deployment decisions. The hybrid nature provides interpretability through symbolic reasoning traces, but neural components remain partially opaque, creating trade-offs between performance and explainability in high-stakes applications.

