# OpenReview forum: "SymRAG: Efficient Neuro-Symbolic Retrieval Through Adaptive Query Routing"
_nesyconf.org/NeSy/2025/Conference_Phase_2 — NeSy 2025 - Phase 2 Poster_

### Official Review · Reviewer_aKxY · 2025-06-28
**SymRAG Review**

**Rating:** 4
**Confidence:** 4

**Review:**

The authors propose SymRAG which introduces adaptive routing for RAG queries to modulate the use of resources.  In this approach, queries are routed to one of three models: neural, symbolic, or hybrid – and this routing depends on aspects of the query and the system state.  The authors implement the system and show it is comparable (appearing slightly better than the baseline) in terms of accuracy while significantly reducing computational overhead.  This idea resonates with the recent interest by some researchers in metacognitive AI – where systems can reason about their own performance and make adjustments accordingly.  I felt the paper did a good job of expressing the novelty of the problem.

However, I had some significant issues with the writing and some of the approach as well.  First, the main part of the paper tells us little about the problem space or algorithm.  Oddly, the authors were not lacking on space (they end the paper at 9.5 pages, and they could have saved a lot of space with compressed versions of Fig 1 and 3.  I felt like I went from some initial definitions to experiments, causing me to read most of the appendix to get an intuition as to the approach, and even that left out a lot of details.

Digging through some of the details, concerned me a bit.  The authors made strong claims about the behavior of the system – that it did not favor the symbolic approach and took the hybrid approach about 60% of the time.  However, the symbolic approach was the fastest, and the weight for the cost was 0.15 (the smallest) – with the weight for expected accuracy set at 0.60.  Likewise, the adaptive threshold algorithm (Algorithm 2) also has hard coded hyperparameters.  Further, these decisions are also linked to quantities of expected accuracy, expected latency, and expected cost – but it is not clear how these are calculated (presumably related to query complexity) – and how dependent are the results on the method used to calculate these quantities or the weights set to them?

Overall, this is an interesting idea, and the authors did obtain some results that seem to validate their hypothesis, but the writing needs to be greatly improved, and the impact of a few key components needs to be reported.

**Anonymity:**

Remain anonymous

---

### Official Review · Reviewer_MT6T · 2025-07-07
**Neuro-Symbolic RAG with Adaptive Query Routing**

**Rating:** 8
**Confidence:** 3

**Review:**

The paper presents SymRAG, a neuro-symbolic framework for RAG that adaptively routes queries to symbolic, neural, or hybrid pipelines. Routing decisions are made in real time, taking into account both the complexity of the query and the current system load. This design enables SymRAG to allocate computational resources more efficiently: simpler queries are quickly handled through symbolic reasoning, while more complex ones are routed to neural or hybrid paths. The approach is evaluated on two complementary QA datasets that emphasize multi-hop and numerical reasoning, using two distinct LLMs. The authors report that SymRAG maintains high answer quality while reducing latency and resource usage, and that disabling its adaptive controller significantly degrades performance.

Strengths
•	Adaptive routing based on query complexity and system load—balancing accuracy and efficiency in real time.
•	Evaluation on two complementary QA datasets and across two different LLMs.
•	Resource utilization is reported in terms of CPU, GPU, and memory consumption.
•	The system implementation is publicly available on GitHub.
•	Extensive supplementary material, including ablation studies and detailed analysis.

The paper is a good fit for the conference. The authors present a well-designed RAG-based system that integrates neural models with symbolic reasoning through an adaptive architecture. SymRAG makes a valuable contribution to ongoing work in neuro-symbolic AI, supported by solid empirical results. It also illustrates how dynamic decision-making can help balance scalability and resource efficiency in real-world applications.

Comments
-	Figure 3 could be moved to the main paper. It provides a clear overview of the system architecture and is especially important because several implementation details—such as the interaction between components and resource management strategies—are not fully described in the main text.

**Anonymity:**

Remain anonymous

---

### Official Review · Reviewer_TVci · 2025-07-10
**Review of SymRAG**

**Rating:** 4
**Confidence:** 3

**Review:**

The authors describe a framework called SymRAG, that uses adaptive query routing to select between symbolic, neural, and hybrid pathways for queries. It adapts based on query characteristics and the system load. This approach is evaluated on 2000 queries across HotpotQA and DROP. The accuracy is competitive (97.6% to 100%), but the resources are utilized a lot more efficiently than without the adaptive approach.

Strengths
1. The submission and the work are relevant to the conference.
2. A useful framework that can use the resources efficiently.

Weaknesses
1. The architecture and the approach have not been discussed in sufficient detail in the main paper. This should be discussed in more detail in the main paper. The architecture diagram is missing from the main paper.
2. Several terms have not been defined. For example, cognitive processing, components in P* and U.
3. Double-blind policy has been violated with the URL of a Github repository.
4. It is not clear how and why LLMs are used.

The results have been summarized multiple times in the paper. Instead of this, the architecture and the approach could have been discussed. Due to these reasons (including the weaknesses), I gave a low score.

**Anonymity:**

Remain anonymous